# Efficient assembly of nanopore reads via highly accurate and intact error correction

Ying Chen[1,14], Fan Nie[2,14], Shang-Qian Xie[3,4,14], Ying-Feng Zheng[1,14], Qi Dai[5,14], Thomas Bray[6], Yao-Xin Wang[5], Jian-Feng Xing[3,4], Zhi-Jian Huang [7,8,9], De-Peng Wang[10], Li-Juan He[1], Feng Luo [11✉], Jian-Xin Wang [2,12✉], Yi-Zhi Liu[1,13✉] & Chuan-Le Xiao [1✉]

Long nanopore reads are advantageous in de novo genome assembly. However, nanopore reads usually have broad error distribution and high-error-rate subsequences. Existing error correction tools cannot correct nanopore reads efficiently and effectively. Most methods trim high-error-rate subsequences during error correction, which reduces both the length of the reads and contiguity of the final assembly. Here, we develop an error correction, and de novo assembly tool designed to overcome complex errors in nanopore reads. We propose an adaptive read selection and two-step progressive method to quickly correct nanopore reads to high accuracy. We introduce a two-stage assembler to utilize the full length of nanopore reads. Our tool achieves superior performance in both error correction and de novo assembling nanopore reads. It requires only 8122 hours to assemble a 35X coverage human genome and achieves a 2.47-fold improvement in NG50. Furthermore, our assembly of the human WERI cell line shows an NG50 of 22 Mbp. The high-quality assembly of nanopore reads can significantly reduce false positives in structure variation detection.

[1] State Key Laboratory of Ophthalmology, Zhongshan Ophthalmic Center, Sun Yat-sen University, #7 Jinsui Road, Tianhe District, Guangzhou, People's Republic of China. [2] School of Information Science and Engineering, Central South University, Changsha 410083, People's Republic of China. [3] Key Laboratory of Genetics and Germplasm Innovation of Tropical Special Forest Trees and Ornamental Plants, Ministry of Education, Hainan University, Haikou 570228, People's Republic of China. [4] Hainan Key Laboratory for Biology of Tropical Ornamental Plant Germplasm, College of Forestry, Hainan University, Haikou 570228, People's Republic of China. [5] College of Life Sciences and Medicine, Zhejiang Sci-Tech University, Hangzhou 310018, People's Republic of China. [6] Oxford Nanopore Technologies, Gosling Building, Edmund Halley Road, Oxford Science Park, Oxford OX4 4DQ, UK. [7] School of Marine Sciences, Sun Yat-sen University, Guangzhou, Guangdong, People's Republic of China. [8] State Key Laboratory of Biocontrol, Sun Yat-sen University, Guangzhou, Guangdong, People's Republic of China. [9] Southern Marine Sciences and Engineering Guangdong Laboratory (Zhuhai), Sun Yat-sen University, Guangzhou, Guangdong, People's Republic of China. [10] Nextomics Biosciences Co., Ltd, Wuhan, People's Republic of China. [11] School of Computing, Clemson University, Clemson, SC 29634-0974, USA. [12] Hunan Provincial Key Lab on Bioinformatics, Central South University, Changsha 410083, China. [13] Research Units of Ocular Development and Regeneration, Chinese Academy of Medical Sciences, Beijing, People's Republic of China. [14] These authors contributed equally: Ying Chen, Fan Nie, Shang-Qian Xie, Ying-Feng Zheng, Qi Dai. ✉email: luofeng@clemson.edu; jxwang@mail.csu.edu.cn; liuyizh@mail.sysu.edu.cn; xiaochuanle@126.com

Reconstructing the genome sequence of a species or individual in a population is one of the most important tasks in genomics[1–3]. Single-molecule sequencing (SMS) technologies, developed by Pacific Bioscience and Oxford Nanopore, yield long reads that can significantly increase the number of solvable repetitive genome regions and improve the contiguity of assembly[4–7]. However, SMS reads usually have high error rates[8]. The two strategies currently used for de novo genome assembly from SMS reads are "correction then assembly" and "assembly then correction." Assemblers, such as Falcon[9], Canu[10], and MECAT[11], first correct SMS reads and then assemble the genome using corrected reads. Conversely, assemblers, such as miniasm[12], Flye[13], wtdbg2[14], Shasta[15], Smartdenovo[16], and Raven[17], assemble the genome using error-prone reads and then correct the assembled genome. Due to the high computational cost of error correction, the "correction then assembly" approach is usually slower than the "assembly then correction" approach. However, directly assembling the genome using error-prone SMS reads can increase assembly errors in the genome sequence, which affects the quality of reference genome and results in bias in downstream analysis, especially in complicated genome regions[10,18]. On the other hand, the "correction then assembly" approach can provide highly continuous and accurate genome assemblies[9–11].

The recently released R9 flow cell from Oxford Nanopore technology can generate reads that are up to 1 M in length and with read N50 > 100 kb, which may significantly improve the contiguity of assembly[5–7,19]. However, errors in nanopore reads are more complex than those in PacBio reads[20,21] (see "Results"). Error correction tools in current assemblers were originally designed for PacBio SMRT (Single Molecule, Real-time) reads and cannot efficiently correct nanopore reads. For example, correcting 30X coverage human nanopore reads using the error correction tool in Canu requires 29 K central process unit (CPU) hours[19]. Moreover, the average identity of reads corrected by Canu is only 92%, which is far less accurate than that of corrected PacBio SMRT reads. These high error rates in corrected nanopore reads can introduce misassemblies. Furthermore, HERS in nanopore reads are usually trimmed during error correction, which reduces both the length of the original reads and contiguity of final assembly.

In this work, we develop NECAT, an error correction and de novo assembly tool designed to overcome the problem of complex errors in nanopore reads. Unlike existing error correction tools that iteratively correct nanopore reads, we develop a two-step progressive method for nanopore read correction. In the first step, NECAT corrects low-error-rate subsequences (LERS), while in the second step, it corrects high-error-rate subsequences (HERS), of the read. This progressive approach allows NECAT to quickly correct nanopore reads, resulting in high accuracy of corrected reads. Evaluated on seven data sets, NECAT is able to obtain low average error rates (1.99–9.01%) and high percentages (45.85–99.34%) of corrected reads having error rate < 5% error, which are both significantly better than the results of Canu. To fully take advantage of nanopore read length, we present a two-stage assembler in NECAT, which constructs contigs using corrected nanopore reads, and then bridges the contigs using original raw reads. We also use an adaptive selection mechanism to choose high-quality supporting reads for each read to be corrected (template read) during error correction, and to select high-quality overlaps for each read during the read-overlap step. Comparing with "correct-then-assemble" assemblers, such as Canu and Canu + Smartdenovo (Canu+S), NECAT is 2.5–258 times faster, while maintaining a similar quality of assemblies. Comparing with "assembly then correction" assemblers, such as Flye, NECAT reports significantly smaller numbers of misassemblies for the assemblies of complex genomes.

## Results

**Analysis of sequencing errors in nanopore reads.** We analyzed sequencing errors in nanopore reads of *E. coli*, *S. cerevisiae*, *A. thaliana*, *D. melanogaster*, *C. reinhardtii*, *O. sativa*, *S. pennellii*, and *H. sapiens* (NA12878) (Supplementary Notes 1–5 and Supplementary Tables 1–2). As shown in Supplementary Table 3, the average error rates of nanopore reads for these eight species ranged from 12.0% (for *S. cerevisiae*) to 20.1% (for *A. thaliana*). Although the average error rates of nanopore reads are similar to those of PacBio SMRT reads, error rates in nanopore reads are more broadly distributed than those of PacBio SMRT reads. The error rates of raw reads in the eight data sets used in our study were broadly distributed between 7 and 50% and centralized between 10 and 20% (Fig. 1A and Supplementary Table 3).

Next, we analyzed sequencing errors in each nanopore read. We first aligned each read onto the reference genomes. We partitioned each read into 500-bp long subsequences and counted the alignment error rate of each subsequence. Our results show that the error rates in each read are also broadly distributed (Fig. 1B). Furthermore, on average, 3–23% of raw reads longer than 10 kb have HERS with error rates > 50% (Supplementary Table 3). Overall, nanopore reads produced by ultra-long library preparation techniques have a higher percentage of reads with

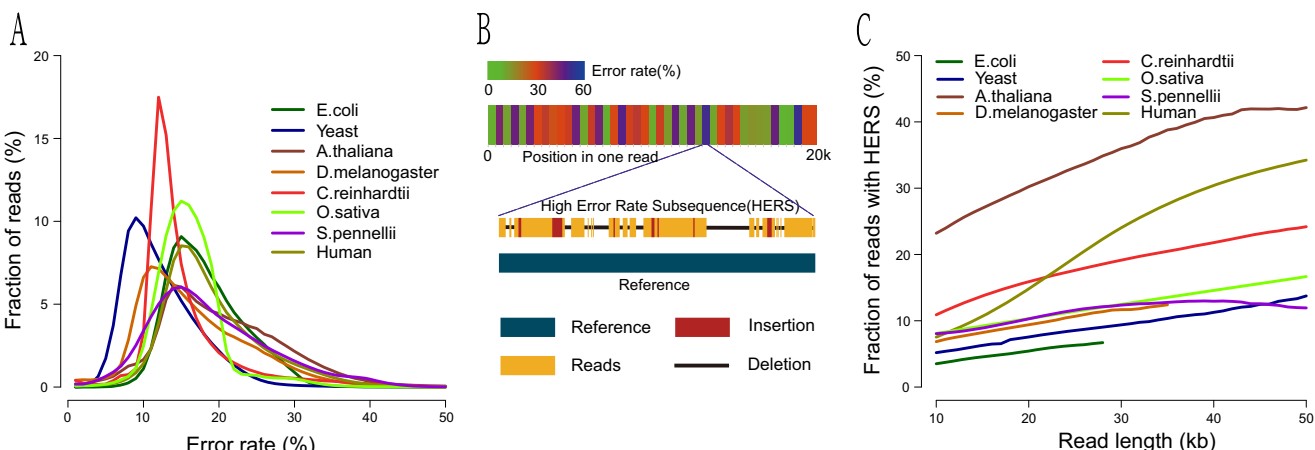

**Fig. 1 Error characteristics of eight nanopore raw read data sets. A** Error rate distribution of raw reads. **B** Error rates of subsequences in a nanopore read (upper) and illustration of a high-error subsequence in the read (bottom). **C** The plot of percentage of raw reads with high-error-rate subsequences (HERS), error rate more than 50% in 500 bp windows against read length.

HERS than those produced by normal library preparation techniques (23 vs. 3–11%). Additionally, the percentage of raw reads with HERS increased as read length increased (Fig. 1C). Especially, for reads produced by ultra-long reads library preparation techniques, up to 45% of raw reads longer than 45 kb have HERS (Fig. 1C). The HERS in nanopore reads usually force the error correction tool to break long reads into shorter fragments, which eliminates the advantage of using long nanopore reads for de novo assembly.

Furthermore, error rates of nanopore reads sampled from different genome locations shared the same distribution except for those of *A. thaliana*, which showed slight variations among genome locations (Supplementary Fig. 1). These results indicated that nanopore sequencing errors did not show genome-location bias. Therefore, a nanopore data set can contain both low- and high-error-rate reads from the same location in a genome.

In summary, our analysis indicates that, unlike PacBio reads, nanopore reads can contain HERS (especially in ultra-long raw reads), and show broad-error-rate distribution among reads and read subsequences.

**Adaptive selection of supporting reads for error correction.** To correct a nanopore read, we first collected supporting reads that overlap with it, then constructed the corrected read using a consensus of multiple sequence alignment of overlapped reads. An overlapping-error-rate threshold is usually set to select supporting reads. Due to the broad distribution of sequencing error rates among nanopore reads, it is difficult to select supporting reads using a single global overlapping-error-rate threshold. Setting a low overlapping-error-rate threshold, such as 0.3 used for PacBio reads, does not generate enough supporting reads to correct nanopore reads with high error rates (>20%); consequently, numerous nanopore reads cannot be corrected. Conversely, setting a high overlapping-error-rate threshold (such as 0.6) results in markedly increasing false supporting reads, which increases computational cost and reduces the accuracy of corrected reads. Furthermore, the high overlapping-error-rate threshold can increase the number of high-error-rate supporting reads for low-error-rate template reads. This results in correcting low-error-rate template with high-error-rate supporting reads, which greatly reduces the accuracy of corrected low-error-rate reads.

To overcome the broad-error-rate distribution of nanopore reads, we used two overlapping-error-rate thresholds to select supporting reads after filtering via distance difference factor (DDF) scoring[11] and k-mer chaining[22] (Online Methods). First, we used a global overlapping-error-rate threshold to select supporting reads. For all template reads, the overlapping error rates of their support reads need to be less than this global threshold. Therefore, the global threshold maintains the overall quality of supporting reads. Based on experiments of error rate distribution of raw reads (Fig. 1A), we set the global overlapping-error-rate threshold to 0.5. Then, for each template read, we set an individual overlapping-error-rate threshold. For each template read, we selected 50 candidate reads with top DDF scores and record the alignment differences between the 50 candidate reads and the template read. Then, the individual overlapping-error-rate threshold is set to be the average of alignment differences minus five times their standard deviation (see "Methods").

The candidate reads were filtered if their alignment error rates were greater than either global or individual overlapping-error-rate thresholds. For low-error-template reads, the individual overlapping-error-rate threshold is less than the global threshold. Conversely, for the high-error-rate template reads, the individual overlapping-error-rate threshold is greater than the global

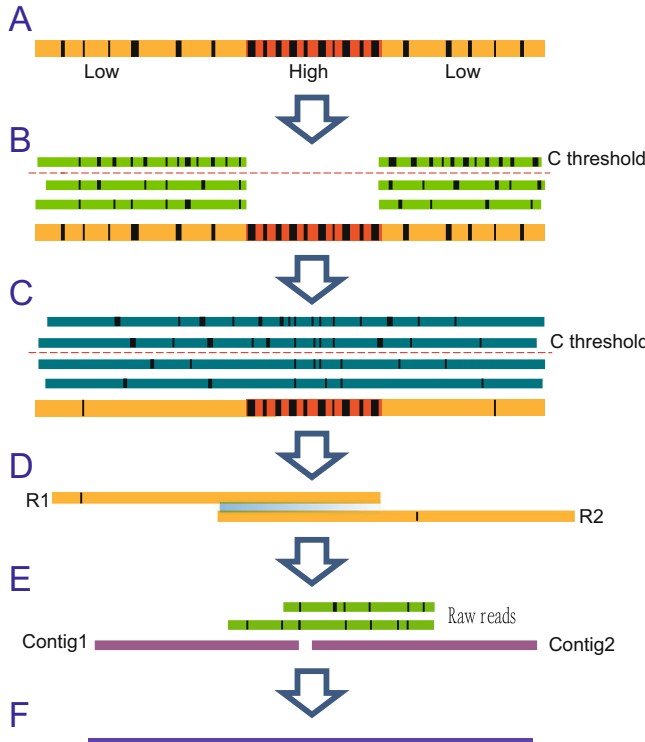

**Fig. 2 Illustration of progressive error correction and two-stage assembly methods of NECAT. A** A template raw read for correction, in which the block boxes indicate errors and the red rectangle is high-error-rate region. **B** Error correction of low-error-rate subsequences. Only low-error-rate subsequences have supporting reads. **C** Error correction of high-error-rate subsequences. **D** Contig assembling using corrected reads. **E** Contig bridging using raw nanopore reads. **F** Output final contigs.

threshold. Using both global and individual overlapping-error-rate thresholds, we were able to maintain the quality of supporting reads for both low and high-error-rate template reads, thereby improving the accuracy of corrected reads. High-error-rate template reads that did not have enough supporting reads were discarded without correction.

**Progressive error correction of nanopore reads.** The supporting reads for error correction are selected according to the average error rate of each template read. Since error rates for subsequences of each nanopore read are also broadly distributed (Fig. 2A), overlapping error rate between supporting reads and HERS can exceed the global overlapping-error-rate threshold 0.5, which can affect the accuracy of corrected subsequences. Therefore, we developed a progressive method for correcting error-prone nanopore reads in two steps (see "Methods"). We first corrected LERS in a template read (Fig. 2B). Then, we corrected HERS (Fig. 2C). In the first step, both corrected and uncorrected subsequences were outputted as a corrected read for the next step. After the first step, we corrected most nanopore reads to high accuracy. This allowed us to obtain an increased number of low-error supporting reads for high-error subsequences in the second step, thereby helping to correct high-error subsequences. After the second step, we outputted only the corrected subsequences. If a subsequence in a template read could not be corrected in the second step, it had either a high error rate or low coverage. Thus, one template read could be broken into multiple corrected reads after the second step.

After filtering the candidate reads using overlapping-error-rate thresholds, we select at most 200 candidate reads with top DDF

scores for each template read. Then, we performed local alignment of candidate reads to the template read (Online Methods). However, performing local alignments of supporting reads to long-template reads is computationally expensive. Although we selected 200 supporting reads for each template read, it was unnecessary to align all these supporting reads when there were enough reads available for error correction. Based on our observation, 12 supporting reads are enough for correcting the template read. Thus, we used a coverage count array (CCA) to record the number of supporting reads that covered each base of the template read during the local alignments. If a template read is covered by 12 support reads, we did not perform local alignments of the rest supporting reads to the template read anymore (see "Methods").

**Progressive assembly of nanopore reads**. The long length of nanopore reads is a significant advantage for de novo genome assembly. However, HERS inside long nanopore reads usually fail to be corrected, leading to the splitting of long nanopore reads into several shorter corrected reads. Using only corrected reads for genome assembly abolishes the advantage presented by the long length of nanopore reads. In this study, we developed a two-step progressive genome assembler for nanopore reads. In the first step, we generated high-quality contigs using the corrected reads (Fig. 2D). In the second step, we bridged the contigs using original nanopore reads to generate final scaffolds (Fig. 2E). The lost contiguity in contigs, caused by HERSs in raw reads, is thereby filled in the second step of the process. Therefore, genome contiguity is improved by maximizing the usage of all raw reads. Our two-step assembly process is similar to the process using SMS reads for scaffolding[23].

Meanwhile, even after error correction, sequencing error rates of corrected nanopore reads (1.5–9%) are still higher than those of corrected PacBio reads (<1%). The reads with high error rates may lead to low-quality overlaps, which have lower identity overlaps, long overhangs, and low-coverage subsequences. The low-quality overlaps increase the difficulty of assembly and introduce errors into assembly results. Therefore, to obtain high-quality contigs, we needed to select high-quality overlaps between corrected reads. Moreover, the error rates of corrected reads also show a relatively broad distribution (Supplementary Note 6 and Supplementary Table 4). Similar to the process used for selecting supporting reads for error correction, we employed both global and individual thresholds to overcome the broad-error-rate distribution for the filtering of low-quality overlaps (see "Methods").

**Performance of NECAT error correction**. We assessed the performance of NECAT error correction using nanopore raw reads of nine data sets: *E. coli, S. cerevisiae, D. melanogaster, A. thaliana, C. reinhardtii, O. sativa, S. pennellii,* NA12878 (rel3,4), and NA12878 (rel6) with respect to correction speed, corrected data size, accuracy and continuity of corrected reads, as well as the number of reads with HERS in corrected reads (Supplementary Note 6). As shown in Table 1, NECAT correction speeds were 2.1–16.5 times faster than those of Canu for nanopore reads of these first seven data sets. The sizes of corrected reads for *E. coli, S. cerevisiae, D. melanogaster, A. thaliana, C. reinhardtii, O. sativa,* and *S. pennellii* were 102.2%, 83.4%, 90.6%, 92.5%, 100.3%, 100.7%, and 91.2% of their raw reads, respectively, while Canu only corrected the longest 40X raw reads and obtained 15.9%, 39.8%, 57.7%, 84.1%, 31.1%, 24.0%, and 28.3% corrected reads from their raw reads, respectively. For NA12878 (rel3,4) and NA12878 (rel6), the sizes of NECAT-corrected reads were 95.1 and 79.5% of their raw reads.

NECAT was able to obtain high-accuracy corrected reads. After the first step, average error rates for *E. coli, S. cerevisiae, D. melanogaster, A. thaliana, C. reinhardtii, O. sativa,* and *S. pennellii* data sets were 4.27%, 3.08%, 7.03%, 11.35%, 4.40%, 6.45%, and 9.23%, respectively; these were less than the average error rates of reads corrected by Canu, which were 7.06%, 3.13%, 8.15%, 12.05, 5.35%, 7.99%, and 9.69%, respectively. After the second step, average error rates for seven data sets were further reduced to 2.23%, 1.53%, 4.89%, 9.01%, 1.99%, 4.66%, and 6.45%, respectively. For NA12878 (rel3,4) and NA12878 (rel6), average error rates of NECAT-corrected reads after the first step were 7.38 and 6.46%, while those after the second step were 7.38 and 6.28%.

The maximum overlapping error rate between corrected reads is usually set to 10% during assembly. Thus, the higher the percentage of corrected reads having <5% error, the more reads can be used for assembly. As shown in Table 1, the percentages of NECAT-corrected reads having error rate < 5% for the first seven data sets were 99.34%, 95.04%, 72.03%, 45.85%, 95.18%, 74.62%, and 63.04%, respectively, which were significantly higher than those of reads corrected by Canu. For two NA12878 data sets, the percentages of NECAT-corrected reads having an error rate of <5% error were 77.60% and 75.45%, respectively.

The progressive correction strategy in NECAT also allowed us to correct more HERS and maintain the contiguity of reads. N50s for NECAT-corrected reads of the first seven data sets were 105.1%, 90.5%, 98.0%, 100.9%, 103.7%, 100.4%, and 96.0%, respectively, of N50s for their corresponding raw reads, indicating that NECAT could preserve the contiguity of raw reads. Conversely, N50s for the reads corrected by Canu were 91.9, 30.4, 85.8, 91.8, 99.0, 97.7, and 87.3% of the corresponding raw reads, which was less than those of NECAT-corrected reads. Another evidence that progressive correction strategy in NECAT can improve the correction of HERS is that the number of reads with HERS has been reduced. After two-step correction using NECAT, the numbers of reads containing HERS in the seven corrected data sets were 1, 268, 3481, 7158, 278, 3511, and 5445, respectively, while Canu-corrected data sets had 1, 4820, 6523, 8722, 726, 4413 and 5511 reads containing HERS. These results indicate that NECAT outperformed Canu in correcting sequencing errors in nanopore raw reads. For two NA12878 data sets, the N50s of NECAT-corrected reads were 106.7% and 121.7% of their raw reads, and the numbers of reads containing HERS were 53,130 and 64,210, respectively.

**Performance of NECAT de novo assembler**. We compared NECAT with two widely used correct-then-assemble pipelines: Canu and Canu + S, as well as an assemble-then-correct pipeline (Flye) for de novo assembly of nanopore reads (Supplementary Note 7). We assembled genomes of *E. coli, S. cerevisiae, A. thaliana, D. melanogaster, C. reinhardtii, O. sativa,* and *S. pennellii* using the longest 40X reads of each data set, and assembled 35X nanopore data for the human NA12878 genome using NECAT only.

As shown in Table 2, NECAT was 6.5–258.2 times faster than Canu, while showing 6.3–577.5 times speedup during the assembly step. Canu employs a high overlapping threshold (14.4%) in its overlapIncore tool for nanopore reads (a low threshold of 6% is used for assembling PacBio reads), which may greatly increase the time cost of local alignments. The Canu + S pipeline replaces the assembly step of Canu with Smartdenovo, which significantly reduces running time. NECAT was still 2.5–57.0 times faster than Canu + S on seven data sets. The high accuracy of corrected reads outputted by NECAT allowed us to use a more rapid overlapping approach. Compared with Flye, NECAT was 1.1–1.8 times faster than Flye on data sets of *A.*

**Table 1 Performance comparison of nanopore read error correction.**

| Data sets | Pipeline | Size (g)/time (h)/ speed (g/h) | Error rate (%) | ≤5%(%) | N50 | N75 | Read number with HERS |
|---|---|---|---|---|---|---|---|
| E. coli | raw reads | 1.38/–/– | 17.8 | 0.01 | 41,074 | 35,484 | 121 |
| | Canu | 0.22/1.63/0.14 | 7.06 | 20.45 | 37,747 | 32,127 | 1 |
| | NECAT | 1.41/0.76/1.86 | 2.23 (4.27) | 99.34 (80.51) | 43,140 | 37,502 | 1 |
| S. cerevisiae | raw reads | 5.48/–/– | 12 | 1.61 | 34,668 | 28,152 | 7589 |
| | Canu | 2.18/30.83/0.071 | 3.13 | 87.3 | 10,554 | 4567 | 4820 |
| | NECAT | 4.57/3.90/1.17 | 1.53 (3.08) | 95.04 (88.09) | 31,364 | 24,480 | 268 |
| D. melanogaster | raw reads | 8.30/–/– | 16.2 | 2.3 | 17,730 | 13,621 | 12,438 |
| | Canu | 4.79/18.10/0.26 | 8.15 | 57.57 | 15,220 | 10,658 | 6523 |
| | NECAT | 7.52/4.20/1.79 | 4.89 (7.03) | 72.03 (64.18) | 17,369 | 13,104 | 3481 |
| A. thaliana | raw reads | 3.08/–/– | 20.1 | 1.57 | 23,386 | 16,253 | 14,483 |
| | Canu | 2.59/12.07/0.22 | 12.05 | 8.09 | 21,472 | 13,133 | 8722 |
| | NECAT | 2.85/1.33/2.14 | 9.01 (11.35) | 45.85 (25.67) | 23,600 | 15,944 | 7158 |
| C. reinhardtii | raw reads | 14.84/–/– | 15 | 1.16 | 54,409 | 46,812 | 4231 |
| | Canu | 4.61/59.40/0.078 | 5.35 | 76.05 | 53,891 | 45,934 | 726 |
| | NECAT | 14.89/11.53/1.29 | 1.99 (4.40) | 95.18 (82.13) | 56,427 | 48,708 | 278 |
| O. sativa | raw reads | 63.40/–/– | 15.6 | 0.49 | 56,325 | 50,847 | 24,205 |
| | Canu | 15.23/43.20/0.35 | 7.99 | 44.42 | 55,010 | 49,612 | 4413 |
| | NECAT | 63.83/18.95/3.37 | 4.66 (6.45) | 74.62 (51.49) | 56,573 | 51,141 | 3511 |
| S. pennellii | raw reads | 132.74/–/– | 18.49 | 1.7 | 24,801 | 22,226 | 127,808 |
| | Canu | 37.53/88.8/0.42 | 9.69 | 34.04 | 21,653 | 19,364 | 5511 |
| | NECAT | 121.07/137.77/0.88 | 6.45 (9.23) | 63.04 (38.77) | 23,810 | 21,480 | 5445 |
| NA12878 (rel3,4) | raw reads | 106.52/–/– | 18.50 | 0.67 | 12,196 | 7209 | 286,641 |
| | NECAT | 101.28/34.65/2.92 | 5.04 (7.38) | 77.60 (34.33) | 13,018 | 7883 | 53,130 |
| NA12878 (rel6) | raw reads | 123.80/–/– | 12.08 | 8.91 | 13,630 | 7984 | 315,117 |
| | NECAT | 98.36/39.35/2.49 | 6.28 (6.46) | 75.45 (77.24) | 14,839 | 9638 | 64,210 |

Size is the total number of base pairs in corrected reads. Time is the running time of correction tools, and the speed is the size/time. Error rate denotes the mean error rate of raw reads and corrected reads; ≤5% denotes the percentage of reads with <5% error rate in total corrected read, values in the bracket are results of NECAT after the first correction; N50 and N75 are the length of reads that reached the 50 and 75% of the total length of all reads; read number with HERS denotes the number of reads that with at least one HERS (more than 50% error in the 500 bp window). The reads used in evaluating the last three metrics (N50, N75, and read number with HERS) of NECAT were corrected from the longest 40× of the raw data set that was selected by Canu by default, see Supplementary Note 6 for details.

thaliana, D. melanogaster, C. reinhardtii, O. sativa, and S. pennellii, and was 224.8 and 21.2 times faster than Flye on data sets of E. coli and S. cerevisiae, but Flye was 3.8 times faster than NECAT on data set NA12878 (rel6).

We then assessed the quality of assembled contigs with respect to assembly size, number of contigs, NG50, NGA50, number of misassemblies, QV, and gene completeness. As shown in Table 2, four assemble pipelines had similar overall performance, while each assembler had its strong points. Canu + S reported a significantly smaller number of contigs for A. thaliana, D. melanogaster, and S. pennellii while NECAT reported a significantly smaller number of contigs for O. sativa and human. NECAT reported significantly higher NG50 for D. melanogaster, O. sativa, and S. pennellii, and Flye reported the higher NG50 for NA12879 (rel6). NECAT reported higher NGA50 for E. coli, O. sativa, S. pennellii, and Flye reported the higher NG50 for NA12879 (rel6). Both NECAT and Flye reported much higher NG50 and NGA50 of NA12878 than those reported by Canu. NECAT reported significantly smaller numbers of misassemblies for both human assemblies, while Canu + S and Flye reported a smaller number of misassemblies for D. melanogaster and Canu reported a significantly smaller number of misassemblies for A. thaliana.

For most species, the four pipelines reported similar values of QV and gene completeness. For O. sativa, the low gene completeness of all four assemblies was due to their low qualities, even after polishing. The Canu reported higher gene completeness for NA12878(rel3,4) (86.3%). Although QVs of polished assemblies were similar, the distributions of error were different. For example, the mismatches and indels per 100 kbp in Canu NA12878 (rel3,4) assembly were 202 and 153, while those in NECAT NA12878 (rel3,4) assembly were 134 and 210.

Next, we assessed the effect of contig bridging in NECAT assembly. As shown in Table 3, the number of contigs was significantly reduced in the assembly of A. thaliana, D. melanogaster, C. reinhardtii, O. sativa, S. pennellii, and H. sapiens genomes after contig bridging of raw reads. For S. cerevisiae, D. melanogaster, C. reinhardtii, S. pennellii, and human NA12878, contig bridging also increased the NG50s and NGA50s of assemblies. These results indicate that contig bridging can improve the contiguity of assembly.

We further compared NECAT assembler with other widely used assemble-then-correct assemblers: miniasm, Smartdenovo, wtdbg2, Flye, Raven, and Shasta (Supplementary Text 1 and Note 7). NECAT has similar time costs as those assemble-then-correct assemblers, but obtains better assembly results, especially for complex genomes (Supplementary Text 1). We also validated our assemblies by comparing them to reference genomes. The qualities of NECAT-generated assemblies were comparable to those of the other correct-then-assemble pipelines and better than assemble-then-correct assemblers (Supplementary Text 2). We also tested hybrid strategies to show the correctness and effectiveness of the error correction and assembly steps of NECAT. These strategies combine either the error correction step of NECAT with the assembly steps of Canu, Smartdenovo, and Flye, respectively, or the error correction step of Canu with the assembly step of Smartdenovo, Flye, and NECAT. The results of the comparison are described in Supplementary Text 3.

**De novo genome assembly of retinoblastoma cell line WERI.** To further evaluate the performance of NECAT in large-genome assembly, we sequenced a cell line called WERI, which is derived from human retinoblastoma[24]. We generated 210 Gb (82-folds) of raw reads from three flowcells using Nanopore PromethION.

**Table 2 The quality and performance of long-read assembly with NECAT.**

| Genome | Pipeline | Assembly size (Mb) | Contig | NG50 (Kb) | NGA50 (Kb) | MA/local MA | QV (pre-/post-polish) | BUSCO | Correct/contig/total time |
|---|---|---|---|---|---|---|---|---|---|
| E. coli | Ref. | 4.6 | 1 | 4642 | – | – | –/– | – | –/–/– |
| | Canu | 4.6 | 1 | 4601 | 3335 | 2/18 | 18.0/22.1 | 18.4% | 26.1/698.1/724.2 |
| | Canu + S | 4.6 | 1 | 4630 | 3287 | 3/2 | 18.6/22.2 | 19.8% | 26.1/8.0/34.1 |
| | Flye | 4.6 | 1 | 4622 | 3071 | 2/2 | 20.2/22.6 | 20.2% | –/–/630.4 |
| | NECAT | 4.6 | 1 | 4595 | 3984 | 2/3 | 18.5/22.3 | 19.8% | 1.6/1.2/2.8 |
| S. cerevisiae | S228C | 12.2 | 17 | 924 | – | – | –/– | – | –/–/– |
| | Canu | 12.7 | 26 | 814 | 703 | 38/33 | 22.3/28.5 | 98.5% | 493.3/1029.9/1523.2 |
| | Canu + S | 12.4 | 19 | 815 | 705 | 34/29 | 22.7/28.9 | 98.2% | 493.3/38.4/531.7 |
| | Flye | 12.3 | 26 | 943 | 706 | 21/26 | 21.8/29.0 | 98.5% | –/–/197.8 |
| | NECAT | 12.3 | 19 | 937 | 708 | 26/35 | 23.1/29.0 | 98.3% | 4.4/4.9/9.3 |
| A. thaliana | TAIR10 | 119.7 | 7 | 23,460 | – | – | –/– | – | –/–/– |
| | Canu | 113.4 | 288 | 6523 | 445 | 478/1152 | 15.6/19.5 | 98.5% | 193.1/1229.9/1423.0 |
| | Canu + S | 115.6 | 44 | 11,071 | 527 | 576/1170 | 15.9/19.6 | 98.8% | 193.1/125.9/319.0 |
| | Flye | 126.6 | 154 | 12,043 | 627 | 1085/1962 | 16.8/18.5 | 98.7% | –/–/59.4 |
| | NECAT | 122.9 | 136 | 11,157 | 582 | 886/1304 | 16.0/18.9 | 98.8% | 19.8/28.0/47.9 |
| D. melanogaster | dm6 | 143.7 | 1870 | 25,287 | – | – | –/– | – | –/–/– |
| | Canu | 146.8 | 499 | 3509 | 3240 | 1307/678 | 20.2/22.2 | 91.3% | 289.6/1259.2/1548.8 |
| | Canu + S | 135.8 | 162 | 14,456 | 6473 | 587/333 | 20.8/23.2 | 91.6% | 289.6/294.4/584.0 |
| | Flye | 139.9 | 593 | 11,925 | 5129 | 558/749 | 21.4/22.5 | 89.9% | –/–/127.9 |
| | NECAT | 143.0 | 277 | 18,072 | 6323 | 1117/1333 | 20.2/22.3 | 92.0% | 37.7/32.7/70.4 |
| C. reinhardtii | Ref. v5.5 | 111.1 | 53 | 7784 | – | – | –/– | – | –/–/– |
| | Canu | 116.4 | 93 | 4564 | 739 | 853/2269 | 19.3/22.2 | 97.9% | 950.4/17,369.6/18,320.0 |
| | Canu + S | 109.7 | 46 | 4498 | 713 | 655/1629 | 20.1/23.0 | 97.7% | 950.4/816.0/1766.4 |
| | Flye | 112.9 | 65 | 6573 | 831 | 764/2029 | 21.6/23.6 | 98.4% | –/–/185.8 |
| | NECAT | 113.4 | 54 | 6169 | 732 | 831/2273 | 19.8/22.4 | 98.0% | 54.8/47.0/101.8 |
| O. sativa | Ref.v4.0 | 382.8 | 15 | 30,829 | – | – | –/– | – | –/–/– |
| | Canu | 383.9 | 385 | 5041 | 2253 | 474/8334 | 15.9/15.9 | 58.6% | 2768.0/16,800.0/19,568.0 |
| | Canu + S | 366.4 | 229 | 3586 | 1832 | 394/5116 | 16.3/16.3 | 59.2% | 2768.0/1926.3/4694.3 |
| | Flye | 380.7 | 249 | 3552 | 2213 | 573/1742 | 16.4/16.3 | 59.2% | –/–/817.6 |
| | NECAT | 373.1 | 120 | 9650 | 3311 | 479/4873 | 16.0/16.3 | 58.4% | 186.9/330.3/517.2 |
| S. pennellii | Ref | 915.6 | 899 | 2522 | – | – | –/– | – | –/–/– |
| | Canu | 961.8 | 2010 | 1664 | 797 | 5614/15,301 | –/20.3 | 97.1% | 5733.1/15,398.4/21,131.5 |
| | Canu + S | 915.6 | 899 | 2522 | – | – | –/– | 97.2% | 5733.1/2510.2/8243.3 |
| | Flye | 1026.0 | 3180 | 1971 | 651 | 8504/10,726 | 16.0/18.5 | 96.7% | –/–/3590.8 |
| | NECAT | 991.8 | 1344 | 4802 | 992 | 5813/12,592 | 15.2/17.3 | 95.5% | 799.6/2434.1/3233.7 |
| NA12878 (rel3,4) | Ref38 | 3272.1 | 639 | 145,139 | – | – | –/– | – | –/–/– |
| | Canu | 2759.0 | 2337 | 5691 | 3368 | 1977/25,179 | 15.4/24.5 | 86.3% | –/–/60,000.0 |
| | NECAT | 2798.4 | 1494 | 14,066 | 9538 | 964/4591 | 16.6/24.6 | 74.9% | 2217.6/5904.0/8121.6 |
| NA12878 (rel6) | Flye | 2867.0 | 3309 | 28,407 | 16,640 | 4054/7258 | 22.9/24.2 | 74.6% | –/–/2500.0 |
| | NECAT | 2846.9 | 1047 | 20,913 | 13,441 | 948/1467 | 23.1/24.4 | 74.5% | 2518.4/6900.4/9418.8 |

"Assembly size" is the total number of base pairs in all contigs generated by assemblers. "NG50" indicates that 50% of reference genome size was contained in contigs having length $\geq n$. "NGA50" is NG50 of aligned blocks that contigs are broken into at mis-assembly breakpoints. "MA/local MA" are the numbers of misassemblies and local misassemblies evaluated by QUAST. "QV" is defined as $10 \times \log_{10}(\frac{100kbp}{\# \text{ mismatches per 100 kbp} + \# \text{ indels per 100 kbp}})$, where "# mismatches per 100 kbp" and "# indels per 100 kbp" are evaluated by QUAST. "BUSCO" is gene completeness evaluated by BUSCO. All the pipelines were tested on the same computer with a 2.0 GHz CPU and 3 T GB RAM of memory. For the first five data sets, we ran all the pipelines on our computer with 32 threads; the correction and contig computational time of the pipelines were recorded. For O. sativa, S. pennellii, and the human data set, we ran all pipelines on our computer with 64 threads, and correction and contig computational time were recorded. The S. pennellii assemblies of Canu and Canu + Smartdenovo were acquired from https://www.plabipd.de/portal/solanum-pennellii, NG50 of which were longer than those generated by us. The S. pennellii assembly of Canu + Smartdenovo was used as the reference genome, and therefore its metrics NGA50, MA, and QV are not evaluated. The NA12878 (rel3,4) assembly and running time of Canu were acquired from public paper. The NA12878 (rel6) assembly and running time of Flye were acquired from https://github.com/fenderglass/Flye.

The WERI genome assembled by NECAT has an N50 of 28 M. To the best of our knowledge, this is the best N50 value for the assembly of the human genome using the general library of the nanopore sequencing platform.

We aligned the WERI assembly to human reference genome hg38 using MUMmer (v4.0)[25]. The dot-plot figure shows that the WERI assembly is structurally consistent with the reference genome except for minor structural variations (Supplementary Note 8 and Supplementary Fig. 2) and the tiling figure shows the continuity of the assembly (Fig. 3). We also used bowtie2[26] to align an Illumina data set for the WERI cell line onto a WERI assembly and hg38 human reference genome. The mapping rate of the WERI assembly (99.1%) was better than that of the hg38 human reference genome (98.0%).

We then identified and validated structural variants (SVs) in the WERI assembly. We detected 11,725 SVs ($\geq$10 bp) in the WERI assembly by aligning it to the hg38 human reference genome using MUMmer (v4.0). We also detected SVs from raw nanopore long reads and Illumina short reads for the WERI cell line using Sniffles[27] and LUMPY[28], respectively (Supplementary Note 8). In all, 7210 SVs are detected using both WERI assembly and raw nanopore reads, while only 1117 SVs are detected using both WERI assembly and NGS (Supplementary Fig. 3 and Supplementary Data 1). Furthermore, we can find 90% of unique

**Table 3 Performance of de novo assemblies before and after the bridging step of NECAT.**

| Species | Stats | Contig | Assembly size (Mb) | Max (Kb) | NG50 (Kb) | MA/local MA | NGA50 (Kb) | QV |
|---|---|---|---|---|---|---|---|---|
| E. coli | Before | 1 | 4.6 | 4587 | 4587 | 2/10 | 3977 | 17.7 |
| | After | 1 | 4.6 | 4595 | 4595 | 2/3 | 3984 | 18.5 |
| S. cerevisiae | Before | 20 | 12.3 | 1530 | 816 | 28/29 | 596 | 22.3 |
| | After | 19 | 12.3 | 1529 | 937 | 26/35 | 708 | 23.1 |
| A. thaliana | Before | 150 | 122.9 | 14,556 | 11,150 | 800/1284 | 535 | 15.7 |
| | After | 136 | 122.9 | 14,567 | 11,157 | 886/1304 | 582 | 16.0 |
| D. melanogaster | Before | 320 | 143.0 | 14,923 | 9612 | 1120/1424 | 4930 | 19.4 |
| | After | 277 | 142.8 | 21,505 | 18,072 | 1117/1333 | 6323 | 20.2 |
| C. reinhardtii | Before | 64 | 113.3 | 8997 | 5515 | 838/2345 | 706 | 19.3 |
| | After | 54 | 113.4 | 9014 | 6169 | 831/2273 | 732 | 19.8 |
| O. sativa | Before | 154 | 372.2 | 22,009 | 9241 | 466/7765 | 3206 | 15.5 |
| | After | 120 | 373.1 | 22,094 | 9650 | 479/4873 | 3311 | 16.0 |
| S. pennellii | Before | 1604 | 991.9 | 22,857 | 3704 | 5762/13,324 | 927 | 15.1 |
| | After | 1344 | 991.8 | 22,879 | 4802 | 5813/12,592 | 992 | 15.2 |
| NA12878 (rel3,4) | Before | 2151 | 2791.6 | 50,857 | 11,980 | 811/6716 | 8334 | 16.0 |
| | After | 1494 | 2798.4 | 73,248 | 14,066 | 964/4591 | 9538 | 16.6 |
| NA12878 (rel6) | Before | 1604 | 2848.6 | 95,968 | 18,488 | 809/1514 | 12,079 | 22.6 |
| | After | 1047 | 2846.9 | 95,975 | 20,913 | 948/1467 | 13,441 | 23.1 |

"Contig" is the total number of contigs in assembly. "Assembly size" is the total number of base pairs in assembly. "Max" is the length of the largest contig. "NG50" indicates that 50% of reference genome size was contained in contigs having length $\geq n$. "NGA50" is NG50 of aligned blocks that contigs are broken into at mis-assembly breakpoints. "MA/local MA" are the numbers of misassemblies and local misassemblies evaluated using QUAST. "QV" is defined as $10 \times \log_{10}(\frac{100kbp}{\text{\# mismatches per 100 kbp} + \text{\# indels per 100 kbp}})$, where "# mismatches per 100 kbp" and "# indels per 100 kbp" are evaluated by QUAST.

small SVs (<1000 bp) that were detected using nanopore raw reads in the WERI assembly, indicating that the assembly can reduce false positives for small SVs (<1000 bp) (Supplementary Data 1).

Next, we examined genes associated with the identified SVs. We found 2843 annotated genes associated with 7210 SVs identified using both WERI assembly and raw nanopore reads. Two hundred and nine of 2843 genes are reported in Phenolyzer[29] and are associated with retinoblastoma (Supplementary Data 2). Among 66 genes, the gene PRKCB, which is scored as high as 0.8901 in Phenolyzer[29], was reported to be involved in retinoblastoma protein phosphorylation[30]. Among the 209 genes, there are eight genes (AATF, PRKCB, PRMT2, FRK, PIK3R1, CUX1, RAC2, and IGF1) with a Phenolyzer score > 0.5, and six of eight genes are associated with retinoblastoma as reported in PubMed. These results indicate that NECAT can provide high-quality assembly for reliable identification of SVs.

## Discussion

Currently, applying nanopore reads in genomic studies is difficult because of the complex errors within these reads. In this study, our analyses have shown that nanopore reads contain HERS, and errors are broadly distributed among nanopore reads and in subsequences of a read. This broad error distribution complicates the selection of supporting reads during the error-correcting process. In traditional error correction methods, the threshold used to select supporting reads can be set too strict or too lenient; the former cannot select enough supporting reads for correction, while the latter generates too many low-quality reads that affect the accuracy of corrected reads. Furthermore, traditional error correction methods cannot correct the high-error subsequences in nanopore reads and generally break nanopore reads into multiple short corrected reads.

In this study, we developed NECAT, which includes methods such as progressive error correction, adaptive supporting reads and alignment selection, and two-stage assembly, to overcome the errors characteristic of nanopore reads. The error correction tool in NECAT, which is 2.1–16.5 times faster than that of Canu, can correct nanopore reads to high accuracy while maintaining the

contiguity of nanopore reads. The whole pipeline is at least 2.5 times faster than other correct-then-assemble pipelines with enhanced or comparable assembly performance. The high performance shown by NECAT suggests that the high error rate of nanopore reads can be overcome by the development of algorithms with respect to error characteristics.

Structural variations identified via raw nanopore reads usually have a high false-positive rate. Here, we show that these false positives can be reduced considerably by using a high-quality assembly of nanopore reads for detection of structural variation. Our results show that NECAT is a useful tool for error correction and assembly of nanopore reads, and for the detection of structural variation.

## Methods

**The architecture of NECAT.** The NECAT pipeline was designed as a high-performance assembler for nanopore reads. To overcome the high error rate of nanopore reads, we developed several methods, including progressive error correction, adaptive supporting reads and read overlaps selection, and two-step assembly. The NECAT pipeline contains four modules (Supplementary Fig. 4): preprocessing, correction, trimming, and assembly. The preprocessing module filters short and ill-formed reads. The correction module uses a progressive strategy to correct nanopore reads in two steps. The trimming module removes low-quality subsequences from corrected reads. The assembly module builds a string graph to assemble the genome in two steps. These four modules can be run in series to finish the assembly or can be operated independently. Currently, NECAT is the most efficient tool for assembling large genomes using nanopore reads.

**Progressive error correction of nanopore reads.** The broad distribution of sequencing error rates among nanopore reads, and within a single nanopore raw read, is the reason why traditional iterative error correction methods usually fail with nanopore data. In this study, we developed a method for correcting nanopore reads. Our progressive error correction method involves two steps. First, we correct the LERS in a read. Then, we correct the HERS in that read using a more sensitive approach. Both steps include the same four sub-steps: (1) selection of candidate reads, (2) determination of alignment-quality threshold, (3) selection of matched reads, and (4) correction of the read. The sub-steps 1, 2, and 4 are the same for both steps. We use different methods to select matched reads for each read to be corrected (template read) in the sub-step 3 of the two steps. In the first step, we use a strict selection method to choose matched reads for the low-error-rate portions of the template read. In the second step, we use a lenient method to choose matched reads for the high-error-rate portions of the template read.

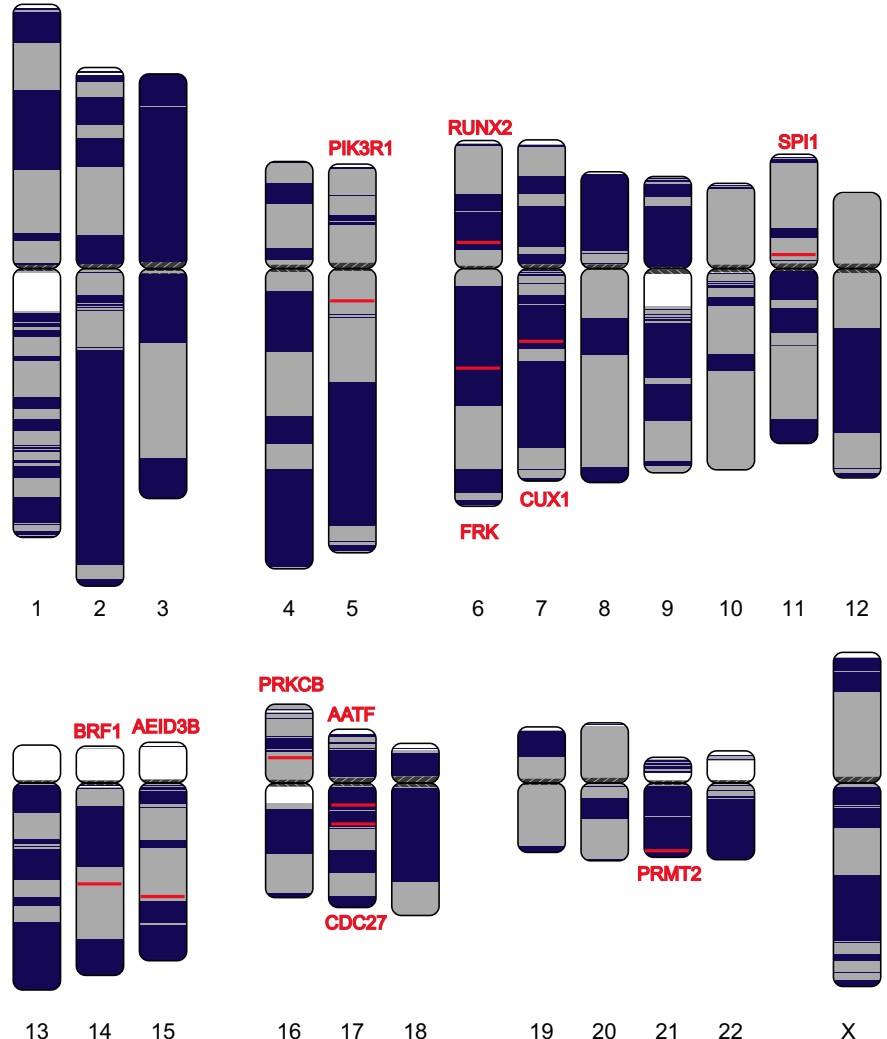

**Fig. 3 Continuity analysis of the assembly of WERI cell line using nanopore reads.** Human chromosomes are painted with assembled contigs using the ColoredChromosomes package. Alternating shades indicate adjacent contigs (each vertical transition from gray to black represents a contig boundary or alignment breakpoint).

**The selection of candidate reads**. For each template read, we selected candidate reads that had overlap with that read. For each pair of reads, we first used the DDF[11] to select a seed k-mer pair with the highest score, which served as a reliable start position for local alignment. However, the wide distribution of error rates decreases the sensitivity of the DDF score for two k-mer pairs that are far apart; this may introduce false positives (Supplementary Fig. 5A). To remove false positives, we gathered all k-mer pairs that support the seed k-pair during DDF scoring. We sorted all k-mer pairs, including the seed k-mer pair, with respect to their positions and then chain them together[22]. The chaining process examines the relative positions of k-mer pairs and helps to filter out false positives (Supplementary Fig. 5B). We then updated the DDF score of the seed k-mer pair with remaining k-mer pairs, which further improved the sensitivity of candidate selection. We recorded the positions of the first and last k-mer pairs in the chain as the approximate mapped positions of candidate read. These two positions, together with the DDF score of the seed k-mer pair, were used for further filtering of redundant candidates and identifying HERS.

**Determination of individual alignment-quality threshold for each template read**. We selected high-quality supporting reads that were used for the correction of each template read. However, broad-error-rate distribution makes it difficult to use a single global threshold for the selection of supporting reads. Besides setting a global overlapping-error-rate threshold to 0.5, we also computed a local individual overlapping-error-rate threshold for each template read. For each template read, we used 50 candidate reads with top DDF scores for local alignments. If a local alignment contained more than 60% of template or candidate read length, we recorded the alignment, and the difference between template and candidate read. If we had $n(0 \le n \le 50)$ recorded alignments and their differences are $d_1$, $d_2$, $\ldots$, $d_n$, we computed their average difference $d_0 = \sum_{i=1}^{n} d_i / n$ and standard deviation

$D = \sqrt{\sum_{i=1}^{n} (d_i - d_0)^2}$. Then, we set the alignment-quality threshold as $d = d_0 - 5D$. This threshold provided a lower alignment-quality bound for low-error-template reads.

**Selection of matched reads**. For each read template, we selected at most 200 candidate reads with top DDF scores for local alignment. We used different alignment methods in the first and second steps. In the first step, we used a blockwise alignment algorithm for aligning supporting reads to the template read. We performed local alignment from the seed k-mer pair in both directions. Thus, we first obtained two semi-global alignments, and then the two alignments were merged into one. Starting from the seed k-mer pair, we partitioned both template and candidate reads into equal-sized blocks 500 bp in length. We then used the Edlib algorithm[31] to successively align each pair of blocks. The aligning process was terminated if the alignment error between a pair of blocks was >50%, or if the alignment algorithm reached the end of a template or candidate read. Because blockwise alignment terminated when either block from template or candidate had a high error rate, we could only obtain alignment between LERS in this step.

In the second step, we used multiple alignment methods to obtain long alignments between templates and candidate reads. We first used the blockwise approach to align the candidate reads to a template. If blockwise alignment terminated early due to the presence of a high-error-rate region inside the template or candidate read, we used the DALIGN algorithm[32] to realign the candidate read to the template. However, alignments produced via DALIGN, running with a large difference threshold of 0.5, were usually too coarse. To refine the alignment result of DALIGN, we then used the Edlib algorithm to perform a global alignment on the mapped subsequences output by DALIGN to get a more correct alignment.

Performing a local alignment of supporting reads to a template is computationally expensive, especially for long-template reads. Usually, only dozens

of alignments are enough for the error correction. Thus, it is unnecessary to align all 200 candidate reads if we have enough supporting reads for error correction. Here, we used a CCA, which is an integer array possessing the same length as that of template read, to record the number of candidate reads that cover each base of the template read. Before aligning a candidate read to the template read, we examined the values of CCA elements between the mapped positions for the approximate start and end of candidate read on a template. If all these values were greater than a user set threshold $C$, we would know that the corresponding region in template read has been covered by enough candidate reads and there was no need to perform the local alignment of this candidate read. If the alignment difference was less than the alignment-quality threshold $d$, we would increase every value of CCA between the start and end template mapped positions by 1. We used a default value of 12 for threshold $C$.

**Correction of nanopore reads.** After selecting the matched candidate reads, we used the FALCON-sense consensus algorithm[9] to correct each subsequence of the template read that was covered by enough candidate reads. In the first step, we replaced these subsequences with corrected subsequences. Then, we output the whole template, including corrected subsequences and uncorrected subsequences, as a corrected read for the next step. HERS was corrected in the next step. In the second step, we only output corrected subsequences, meaning that one template may produce more than one corrected read. If a subsequence in a template read was not able to be corrected in the second step, it either had a too high error rate or low coverage.

**Trimming of low-quality subsequences.** Long nanopore reads may still contain HERS even after error correction, which can greatly affect the quality of assembly. Thus, low-quality subsequences need to be trimmed before assembly. We only selected 40X coverage longest corrected reads for trimming and future assembly. First, we performed pairwise alignment on selected nanopore reads using the trimming module of MECAT[11]. Because even corrected nanopore reads could have a relatively high error rate, we used the sensitive DALIGN algorithm to replace the original diff algorithm in the MECAT trimming module before performing local alignments. After pairwise alignment, we gathered high-quality overlaps with more than 90% identity for each read. If every residue of a read was covered by at least one overlap, the read was designated as a complete read. On the other hand, if there were subsequences without overlap coverage in a read, we trimmed it to its longest covered subsequence, which was called a trimmed read.

After trimming, the reads were usually subjected to another pairwise alignment. Our experiments showed that <10% of corrected reads were trimmed, therefore, it was unnecessary to pairwise align 90% of untrimmed reads. Thus, we stored complete reads and trimmed reads separately after trimming. Pairwise alignments were only performed between complete reads and trimmed reads, and between trimmed reads. The results of these pairwise alignments, together with complete reads, trimmed reads, and results of original pairwise alignments between complete reads, were fed into the assembly module.

**De novo assembly of nanopore reads.** Although the long length of nanopore reads helps improve genome assembly, the relatively high error rate of these reads rendered genome assembly difficult. Here, we developed an assembly tool, which is particularly useful for nanopore reads because it can overcome the high error rate of these reads. Our assembly module in NECAT consists of three steps: filtering of low-quality read overlaps, contig assembly, and contig bridging. We used multiple quality-control measures to filter out low-quality overlaps between nanopore reads. Then, we constructed a directed string graph and solved the graph to generate contigs. Finally, we bridged the contigs using original reads to generate the final scaffolds.

**Filtering of low-quality read overlaps.** Low-quality overlaps complicate assembly and introduce errors into assembly results. In NECAT, we used multiple thresholds to control the identity, overhang, and coverage of overlaps to filter out low-quality overlaps. For each read, we determined the coverage of each base according to its overlaps. Then, we calculated the minimum coverage ($c_{min}$), maximum coverage ($c_{max}$) of bases, as well as the difference between minimum coverage and maximum coverage ($c_{diff}$). If its $c_{min}$ was less than the predefined threshold, min_coverage, or $c_{max}$ was larger than the predefined threshold, max_coverage, or $c_{diff}$ was larger than the predefined threshold, max_diff_coverage, the read and its overlaps were removed. The details on coverage threshold settings are provided in Supplementary Note 9. Because of broad error distribution among different reads, we used both global and local threshold, instead of a single global threshold, for quality control of overlap identity and overhang. For a high-quality read, the average quality of its overlaps was higher than the global average; therefore, we set the local threshold to filter out overlaps having relatively low quality. For a low-quality read, the average quality of its overlaps was lower than the global average; we then used the global threshold to filter out low-quality overlaps for that read. This strategy allowed us to filter out overlaps with relatively low quality for each read, and to maintain the overall quality of all the overlaps. Details on setting global and local thresholds for overlap identity and overhang are provided in Supplementary Note 9.

**Contig assembly.** Next, we constructed a directed string graph and removed transitive edges using Mayer's algorithm[33]. We marked the best out-edge and the best in-edge of each node based on overlap lengths of the edges. The edges that were not marked as best out-edge or best in-edge were removed[34]. We also removed ambiguous edges (tips, bubbles, and spurious links) in the graph. We then identified linear paths from the graph and generate contigs. When there was a branch, we broke the path to generating multiple contigs, which could reduce the possibility of mis-assembly.

**Contig bridging.** During error correction, long reads with high-error subsequences were cut into multiple shorter reads, which eventually led to the discontinuity of contigs. It is possible to relink contigs using long raw reads[23,35]. First, we aligned the long raw reads to contigs. Two contigs could have an overlap that had low quality; this overlap was filtered before the construction of a string graph. A raw read could either fill the gap between two contigs, which was then called a gap read, or overlap with the overlap of two contigs, which was then called an overlapped read. For each raw read, we recorded the gap or overlap length between the mapped positions on the ends of the two contigs. For each pair of contigs, the raw reads connecting them were grouped as those connecting in the same orientation or those connecting in different orientations. In each orientation group, we clustered the raw reads based on their gap/overlap lengths. If the difference between the gap/overlap lengths of two raw reads was less than a threshold (default value is 1000 bp), we assigned them into the same cluster. And we assigned a score to each raw read, which was the sum of the products of identity and length of overlaps between the raw read and the pair of contigs. The read cluster with the largest sum of scores was chosen as the link for the contig pair.

After identifying links between contig pairs, we created a string graph in which contigs were nodes, and links between the contigs were edges. The weight of each edge was set to the link score. We simplified the graph again by removing transitive edges. Then, we traversed the graph and identified linear paths as the final contigs. A raw read from the link was selected to fill the gap between contigs.

**Error distribution analysis.** We analyzed error distribution in nanopore data sets for *E. coli, S. cerevisiae, A. thaliana, D.* melanogaster, *C. reinhardtii, O. sativa,* and *S. pennellii*. Our results indicated that the sequencing error rate of nanopore reads was high at 10–20%, which helped us to refine our algorithm for the NECAT platform and provided insights into why the existing correction algorithms were not suitable for the correction of nanopore reads. Details are provided in Supplementary Note 5.

**Evaluation.** We compared our error correction tool with those provided in Canu. We also systematically evaluated the assembly tools provided in NECAT by comparing them with those of Canu, Canu + S, and Flye. Details of these comparisons are reported in Supplementary Notes 6, 7, and 10.

**Reporting summary.** Further information on research design is available in the Nature Research Reporting Summary linked to this article.

## Data availability

We used nine data sets to evaluate the performance of NECAT. Among these data sets, those for *Saccharomyces cerevisiae, Oryza sativa, and Homo sapiens* (the WERI human retinoblastoma cell line) were generated using our in-house sequencing, while the other four were obtained from public websites. The details on the data used in this study are reported in Supplementary Notes 1–4. All data that support the findings of this study are available on https://github.com/xiaochuanle/necat and http://www.tgsbioinformatics.com/necat/. The nanopore whole-genome sequencing data sets of *Saccharomyces cerevisiae, Chlamydomonas reinhardtii, Oryza sativa,* and Human retinoblastoma cell line WERI from this study have been deposited in the Genome Sequence Archive of BIG Data Center, Beijing Institute of Genomics (BIG, http://gsa.big.ac.cn), Chinese Academy of Sciences, with Project Accession No. "PRJCA003787" and GSA Accession Nos. "CRA003449," "CRA003450," and "HRA000410."

## Code availability

All codes that support the findings of this study are available on https://github.com/xiaochuanle/necat and http://www.tgsbioinformatics.com/necat/.

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

## Acknowledgements

We thank all those who generated and freely released the data analyzed in our present study. This study was funded in part by the National Natural Science Foundation of China (Grant Numbers. 31871326, 31701146, 91953122, 61732009, U1909208, 81530028, and 81721003) and Guangdong Basic and Applied Basic Research Foundation (2020B1515020057). We thank the Local Innovative and Research Teams Project of Guangdong Pearl River Talents Program, CAMS Innovation Fund for Medical Sciences (2019-I2M-5-005), Clinical Innovation Research Program of Guangzhou Regenerative Medicine and Health Guangdong Laboratory (Grant Number 2018GZR0201001); the State Key Laboratory of Ophthalmology, Zhongshan Ophthalmic Center, Sun Yat-sen University. This work was supported in part by the US National Institute of Food and Agriculture (NIFA; Grant Number 2017-70016-26051) and US National Science Foundation (NSF; Grant Number ABI-1759856) to F.L.

## Author contributions

C.L.X., Y.Z.L., J.X.W., and F.L. conceived and designed this project. Y.C. and C.L.X. conceived, designed, and implemented the consensus algorithm. F.N. and C.L.X conceived, designed, and implemented the progressive assembly algorithm. F.N. and Y.C. integrated all the programs into the NECAT pipeline and provided documentation. S.Q.X., C.L.X., Y.X.W., J.F.X., and Q.D. ran analyzed genome assemblies and analyzed the performance of algorithms developed in this study. T.B., Z.J.H., D.P.W., and L.J.H. coordinated data release and assisted with executing the pipeline. F.L., Y.C., and F.N. performed the theoretical analysis of the algorithms developed in this study. F.L., Y.C., F.N., S.Q.X., Y.F.Z., and C.L.X. wrote the paper. All authors have read and approved the final version of this paper.

## Competing interests

The authors declare no competing interests.
