## [Peer Review File · Nature Communications]

REVIEWER COMMENTS

Reviewer #2 (Remarks to the Author):

Dear authors,

I like the work in the manuscript by Chen et al. Their error correction scheme is novel, and the assembly results are promising. I have some comments, listed below:

1. What are the basecaller versions for the data used in these experiments? important to include dataset details.
2. 10-30% error rates seem high. 30% data almost suggests many year old nanopore data. Is that correct? It would be good to include the true distribution of errors.
3. Since this is an assembly paper, I found it odd that comparisons to Shasta or Raven weren't performed? I think that comparison would be useful.
4. Figure 2 has unlabeled elements, which make interpretation difficult. What do the black boxes in 2A correspond to? Is 2A showing a template read? Every rectangle should be unambiguous.
5. In all assembly analysis, the information on misassemblies was missing (even though misassemblies are mentioned in the online methods). This is an important metric for an assembler.
6. It is worth performing gene completeness analysis, such as BUSCO.
7. Could the authors comment on the effect of running error correction using the proposed strategy and then assembling with other tools using the corrected reads? And vice versa?
8. No information on error correction was shown for human dataset?
9. Some of the datasets used in the dataset are old, e.g. the NA12878 data used was over 2 years old and has been updated. I suggest the authors perform these analyses with updated data.
10. I appreciate that the authors saw hither errors in longer reads, but the explanation on how they were calculated was not clear. This section could use a bit of clarity in the text.
11. I noted that a lot of emphasis was given to N50s for assemblies, but there is no discussion of NG50s and more importantly, NGA50s. It is important to mention those. The authors could run QUAST to do this.

I send my best wishes to the authors.

Generally, there are a few errors in the text that could be resolved with a careful proofreading for grammar, language, and typos.

Reviewer #3 (Remarks to the Author):

In their paper "Efficient assembly of Nanopore reads via highly accurate and intact error correction", Chen et al. present a new approach for correction and assembly of ONP long reads. ONP reads are more challenging to assemble than PacBio CLR reads and the authors use clever algorithmic tricks (e.g. adaptive error threshold for alignment filtering) to circumvent those challenges. I was able to run the software on two datasets that the authors cite and I could reproduce their results. I like their approach but I have three major concerns/suggestions:

1) The authors use very superficial and vague descriptions in the main text to describe their approach. I understand that many details are in the Online Methods but this is a methods paper and sufficient details should be presented in an objective style in the main body of the paper so that the readers can follow the method. I'll give some examples (but this is not an exhaustive list).

Line 95: "for each template read" - Template read is not defined in the manuscript. Is this the read that all reads are being aligned to?

Line 150: "global overlapping-error-rate threshold" - How is this threshold determined?

Line 151 "overall quality of supporting reads" - what do the authors mean by quality in this particular context?

Line 152 "we set an individual overlapping-error-rate threshold" - How is this threshold determined?

Line 178 "Usually, twelve supporting reads are enough for error correction" - Is this an observation that the authors made or is it known for ONP reads in general?

Line 184: "For template read covered by a sufficient number of support reads, we did not perform local alignment of supporting reads to this region anymore" - Do they mean that reads above a coverage threshold were not considered? How is this coverage determined?

Line 204: "we needed to select high-quality overlaps between corrected reads because low-quality overlaps" what is high quality and low quality here?

The continuity of the corrected is reported to be >100% of the raw reads. How is that possible?

2) The authors should include assembly metrics from Flye in Table 2 and perhaps remove Canu (but keep Canu+smartdenovo). Inclusion of Flye is necessary and fair because the authors begin their paper by citing assemblers of two categories ("correction then assembly" and "assembly then correction") and in Table 2 they present their results along with results from Canu and Canu+smartdenovo, both of which belong to the first category.

3) The authors should include quality estimates of their assemblies using Illumina consensus QV and metrics like NGA50, which takes assembly errors into account while calculating N50 (e.g. they can use QAST + GAGE against the reference genomes). Right now, there is no way to tell whether the NECAT assemblies have more errors than the canu+smartdenovo or Flye. And the authors should calculate these quality metrics for both before and after bridging and compare them with Canu (or Canu+smartdenovo) and Flye.

Some minor comments:

1) The bridging approach is very similar to the concept of contig-bridging implemented in finisherSc (<https://academic.oup.com/bioinformatics/article/31/19/3207/210799>). I think the authors should acknowledge the existence of finisherSc when they talk about bridging.

2) The argument that ONP long reads are superior to shorter PacBio reads is not very objective. As theory would predict and as the HiCanu paper (<https://www.biorxiv.org/content/10.1101/2020.03.14.992248v3>) shows, you can get extremely contiguous human genomes (far more than achievable by similar coverage ONP) from shorter but accurate PacBio reads.

Summary

We appreciate the valuable comments and suggestions from the editor and reviewers. Based on the suggestions and comments from editor and reviewers, we revised our paper. We addressed those comments and suggestions carefully and included a point-by-point response below. We completely rewrote some paragraph and significant changes were highlighted by red color.

Response to Reviewer #2 (Remarks to the Author):

1. What are the basecaller versions for the data used in these experiments? important to include dataset details.

The basecaller versions were given in the revised Supplementary Note 3 now, some detail information of dataset was included in Supplementary Note 1-2.

2. 10-30% error rates seem high. 30% data almost suggests many year old nanopore data. Is that correct? It would be good to include the true distribution of errors.

This is a typo. The “10-30% error rates” were corrected to “10-20% error rates”. This corrected number can be seen from supplementary table 3 and Figure 1A.

3. Since this is an assembly paper, I found it odd that comparisons to Shasta or Raven weren't performed? I think that comparison would be useful.

We assembled the datasets of seven genomes using Shasta and Raven and compared the results with those of other pipelines. The metrics of assemblies were added to the revised Supplementary Table 7. The results were discussed in the revised Supplementary Text 1 and 2.

4. Figure 2 has unlabeled elements, which make interpretation difficult. What do the black boxes in 2A correspond to? Is 2A showing a template read? Every rectangle should be unambiguous.

We added descriptions to explain the ambiguous elements in the revised Figure 2.

5. In all assembly analysis, the information on misassemblies was missing (even though misassemblies are mentioned in the online methods). This is an important metric for an assembler.

According to your suggestion, we evaluated the assemblies using QAST and the metrics were recorded in Table 2, Table 3, Supplementary Table 7 and Supplementary Table 11.

6. It is worth performing gene completeness analysis, such as BUSCO.

According to your suggestion, we performed gene completeness analysis using BUSCO. The metrics were recorded in the revised Table 2 and the revised Supplementary Table 7.

7. Could the authors comment on the effect of running error correction using the proposed strategy and then assembling with other tools using the corrected reads? And vice versa?

We tested hybrid strategies which combine the correction step of NECAT with the assembly steps of other tools, or the correction step of Canu with the assembly step of NECAT. We used those hybrid assembly pipelines to assemble datasets of *E. coli*, *S. cerevisiae*, *A. thaliana*, *D. melanogaster* and *C. reinhardtii*. The details were given in the revised Supplementary Text 3 and the metrics of the assemblies were shown in the revised Supplementary Table 11. Our comparison showed that NECAT reported consistent performance on the assemblies of all five genomes, while the performances of other hybrid pipelines were not stable on all five genomes.

8. No information on error correction was shown for human dataset?

We added the performance of NECAT error correction for two human NA12878 datasets to Table 1 and Supplementary Table 4. Due to limitation computational resources, we could not obtain the performance of Canu error correction for two human NA12878 datasets in limited time.

9. Some of the datasets used in the dataset are old, e.g. the NA12878 data used was over 2 years old and has been updated. I suggest the authors perform these analyses with updated data.

We assembled the newer NA12878 (rel6) data and the results were reported in the revised Table 2-3.

10. I appreciate that the authors saw hither errors in longer reads, but the explanation on how they were calculated was not clear. This section could use a bit of clarity in the text.

We updated the description of how we counted the error rate in the subsequences of each reads in the second paragraph of the Section “Analysis of sequencing errors in Nanopore reads”.

11. I noted that a lot of emphasis was given to N50s for assemblies, but there is no discussion of NG50s and more importantly, NGA50s. It is important to mention those. The authors could run QUAST to do this.

We added the NGA50s to evaluation metrics. The corresponding results were given in the revised Table 2-3, Supplementary Table 7 and Supplementary Table 11.

Generally, there are a few errors in the text that could be resolved with a careful proofreading for grammar, language, and typos.

In revised version, we corrected all errors and typos and carefully edited our paper.

Response to Reviewer #3 (Remarks to the Author):

In their paper "Efficient assembly of Nanopore reads via highly accurate and intact error correction", Chen et al. present a new approach for correction and assembly of ONP long reads. ONP reads are more challenging to assemble than PacBio CLR reads and the authors use clever algorithmic tricks (e.g. adaptive error threshold for alignment filtering) to circumvent those challenges. I was able to run the software on two datasets that the authors cite and I could reproduce their results. I like their approach but I have three major concerns/suggestions:

1) The authors use very superficial and vague descriptions in the main text to describe their approach. I understand that many details are in the Online Methods but this is a methods paper and sufficient details should be presented in an objective style in the main body of the paper so that the readers can follow the method. I'll give some examples (but this is not an exhaustive list).

Line 95: "for each template read" - Template read is not defined in the manuscript. Is this the read that all reads are being aligned to?

Line 150: "global overlapping-error-rate threshold" - How is this threshold determined?

Line 151 "overall quality of supporting reads" - what do the authors mean by quality in this particular context?

Line 152 "we set an individual overlapping-error-rate threshold" - How is this threshold determined?

Line 178 "Usually, twelve supporting reads are enough for error correction" - Is this an observation that the authors made or is it known for ONP reads in general?

Line 184: "For template read covered by a sufficient number of support reads, we did not perform local alignment of supporting reads to this region anymore" - Do they mean that reads above a coverage threshold were not considered? How is this coverage determined?

Line 204: "we needed to select high-quality overlaps between corrected reads because low-quality overlaps" what is high quality and low quality here? The continuity of the corrected is reported to be >100% of the raw reads. How is that possible?

In revised version, we added the more descriptions in the text according to the suggestions.

2) The authors should include assembly metrics from Flye in Table 2 and perhaps remove Canu (but keep Canu+smartdenovo). Inclusion of Flye is necessary and fair because the authors begin their paper by citing assemblers of two categories ("correction then assembly" and "assembly then correction") and in Table 2 they present their results along with results from Canu and Canu+smartdenovo, both of which belong to the first category.

In the revised version, we added the Flye to our comparison according to the suggestion. We compared NECAT to Canu, Canu+smartdenovo and Flye in Table 2. We also compared to other assembly then correction pipelines in supplementary.

3) The authors should include quality estimates of their assemblies using Illumina consensus QV and metrics like NGA50, which takes assembly errors into account while calculating N50 (e.g. they can use QUAST + GAGE against the reference genomes). Right now, there is no way to tell whether the NECAT assemblies have more errors than the canu+smartdenovo or Flye. And the authors should calculate these quality metrics for both before and after bridging and compare them with Canu (or Canu+smartdenovo) and Flye.

According to your suggestion, we added new metrics like QV, NGA50, and gene completeness to evaluate the assemblies. We also added new metrics to evaluate the assemblies before and after bridging. The corresponding results were given in Table 2, Table 3, Supplementary Table 7 and Supplementary Table 11.

Some minor comments:

1)The bridging approach is very similar to the concept of contig-bridging implemented in finisherSc(<https://academic.oup.com/bioinformatics/article/31/19/3207/210799>). I think the authors should acknowledge the existence of finisherSc when they talk about bridging.

According to your suggestion, we cited the finisherSc paper in the Method as Ref 35.

2) The argument that ONP long reads are superior to shorter PacBio reads is not very objective. As theory would predict and as the HiCanu paper(<https://www.biorxiv.org/content/10.1101/2020.03.14.992248v3>) shows, you can get extremely contiguous human genomes (far more than achievable by similar coverage ONP) from shorter but accurate PacBio reads.

According to your suggestion, we updated the argument as “The recently released R9 flow cell from Oxford Nanopore technology can generate reads that are up to 1M in length and with read N50 >100 kb, which may significantly improve the contiguity of assembly^{5-7, 19}.”

REVIEWERS' COMMENTS

Reviewer #2 (Remarks to the Author):

Dear authors,

Thank you for addressing reviewer comments and making the changes. The manuscript looks better now.

I do not have any further comments, and send my best wishes.

Reviewer #3 (Remarks to the Author):

I would like to thank the authors for their detailed responses to my comments. The manuscript reads much better and is more clear than before. However, in comparison to the results from Flye, which is now included in the main Tables and text, NECAT does not seem to provide any major advantage with its new error correction and assembly approach. Having said that, the Nanopore community might still find NECAT useful. I only have some minor comments -

Please check grammar in this sentence at line 136-137

"majority of Nanopore reads results in markedly increasing false supporting reads"

At Line 147-148, the authors state "Based on experiments, we set the global overlapping-error-rate threshold to 0.5"

what kind of experiments? A single sentence explaining the experiment should suffice.

Line 149-150: the authors say "For each template read, we selected 50 candidate reads with top DDF scores"

Does this criteria assume a minimum read coverage for the whole dataset? Do you need a the minimum overall read coverage to satisfy this criteria?

Line 341-342: "human reference genome using Nummer (v4.0)"

Do the authors mean "nucmer" or "MUMmer"?

Line 344-345: "7210 SVs are commonly detected using WERI assembly"

'commonly' is not the right word here. Please revise this sentence. Same goes for 'commonly' in the next sentence.

Line 347-348: "Furthermore, 90% of unique small SVs (<1000 bp) detected using Nanopore raw reads were able to be found in the WERI assembly"

Please check this sentence.

Summary

We appreciate the valuable comments and suggestions from the editor and reviewers. Based on the suggestions and comments from editor and reviewers, we revised our paper. We addressed those comments and suggestions carefully and included a point-by-point response below.

Response to Reviewer #3 (Remarks to the Author):

I would like to thank the authors for their detailed responses to my comments. The manuscript reads much better and is more clear than before. However, in comparison to the results from Flye, which is now included in the main Tables and text, NECAT does not seem to provide any major advantage with its new error correction and assembly approach. Having said that, the Nanopore community might still find NECAT useful. I only have some minor comments

Comparing to “assembly then correction” approached used in Flye, the “correction then assembly” has the advantage on assembly of complex genomes, which will lead to less assembly errors. For examples, the NECAT has reported significantly smaller numbers of misassemblies for both human assemblies. Correct assembly error is a time-consuming task.

Please check grammar in this sentence at line 136-137

"majority of Nanopore reads results in markedly increasing false supporting reads"

We correct the sentence to solve the confusion: “setting a high overlapping-error-rate threshold (such as 0.6) results in markedly increasing false supporting reads”.

At Line 147-148, the authors state "Based on experiments, we set the global overlapping-error-rate threshold to 0.5"

what kind of experiments? A single sentence explaining the experiment should suffice.

In the revised version, we modified the sentence to make it clearer. “Based on experiments of error rate distribution of raw reads (Figure 1 A), we set the global overlapping-error-rate threshold to 0.5.”

Line 149-150: the authors say "For each template read, we selected 50 candidate reads with top DDF scores"

Does this criteria assume a minimum read coverage for the whole dataset? Do you need a the minimum overall read coverage to satisfy this criteria?

The 50 is used to cap the maximum number of candidate reads. We do not set a coverage criterion on the raw read dataset. If a template has more than 50 candidate reads, we then choose the 50 candidate reads with top DDF scores. Otherwise all the candidate reads

will be used. When a template read has less than 4 candidate reads, it cannot be corrected. So, the minimum overall read coverage is 4.

Line 341-342: "human reference genome using Nummer (v4.0)"

We used the MuMmer. We correct the typo in the revision

**Line 344-345: "7210 SVs are commonly detected using WERI assembly"
'commonly' is not the right word here. Please revise this sentence. Same goes for
'commonly' in the next sentence.**

We correct the sentence as following: "7210 SVs are both detected using WERI assembly and raw Nanopore reads, while only 1117 SVs are both detected using WERI assembly and NGS".

Line 347-348: "Furthermore, 90% of unique small SVs (<1000 bp) detected using Nanopore raw reads were able to be found in the WERI assembly". Please check this sentence.

We modify the sentence to make it clearer: "Furthermore, we can find 90% of unique small SVs (<1000 bp) that were detected using Nanopore raw reads in the WERI assembly".